# Global River Radar Altimetry Time Series (GRRATS): New River Elevation Earth Science Data Records for the Hydrologic Community

Stephen Coss[1,2], Michael Durand[1,2], Yuchan Yi[1], Yuanyuan Jia[1], Qi Guo[1], Stephen Tuozzolo[1], CK Shum[1,6], George H. Allen[3,4], Stéphane Calmant[5], Tamlin Pavelsky[3]

[1] School of Earth Sciences, The Ohio State University, Columbus, Ohio, USA

[2] Byrd Polar and Climate Research Center, The Ohio State University, Columbus, Ohio, USA

[3] Department of Geological Sciences, The University of North Carolina at Chapel Hill, USA

[4] Department of Geography, Texas A&M University, College Station, TX, USA

[5] IRD/LEGOS, 16 Avenue Edouard Belin, 31400 Toulouse, France

[6] Institute of Geodesy and Geophysics, Chinese Academy of Sciences, Wuhan 430077, China

*Correspondence to*: Stephen Coss (coss.31@osu.edu)

**Abstract.** The capabilities of radar altimetry to measure inland water bodies are well established and several river altimetry datasets are available. Here we produced a globally-distributed dataset, the Global River Radar Altimeter Time Series (GRRATS), using Envisat and Ocean Surface Topography Mission (OSTM)/Jason-2 radar altimeter data spanning the time period 2002–2016. We developed a method that runs unsupervised, without requiring parameterization at the measurement location, dubbed virtual station (VS) level and applied it to all altimeter crossings of ocean draining rivers with widths >900 m (>34% of global drainage area). We evaluated every VS, either quantitatively for VS locations where in-situ gages are available, or qualitatively using a grade system. We processed nearly 1.5 million altimeter measurements from 1,478 VS. After quality control, the final product contained 810,403 measurements distributed over 932 VS located on 39 rivers. Available in-situ data allowed quantitative evaluation of 389 VS on 12 rivers. Median standard deviation of river elevation error is 0.93 m, Nash-Sutcliffe efficiency is 0.75, and correlation coefficient is 0.9. GRRATS is a consistent, well-documented dataset with a user-friendly data visualization portal, freely available for use by the global scientific community. Data are available at DOI 10.5067/PSGRA-SA2V1(Durand et al., 2016).

# 1 Introduction

Despite growing demand from emerging large-scale hydrologic science and applications, global and freely available observations of river water levels are still scarce (Hannah et al., 2011; Pavelsky et al., 2014; Shiklomanov et al., 2002). Advances in remote sensing and computing capabilities have enabled new areas of global fluvial research that are dependent upon river elevations, including global hydrologic quantification of carbon and nitrogen fluxes (e.g. Allen & Pavelsky, 2018; Oki & Yasuoka, 2008) and characterizing flood risk for future climate scenarios (Schumann et al., 2018; Smith et al., 2015). Evaluation of these global river elevation models requires global datasets of river elevation time series, but in situ river water levels are scarce, as they are often not shared outside specific government agencies. Thus model evaluation and calibration increasingly relies on remotely sensed data (Overton, 2015; Pavelsky et al., 2014; Sampson et al., 2015). Newer radar altimeter missions like Sentinel-3 are improving the contemporary record with features like automated processing, alleviating the need for retracking and other post processing to generate useful measurements. In addition, the Surface Water and Ocean Topography (SWOT; swot.jpl.nasa.gov) satellite mission, scheduled for launch in 2021, will observe global river elevations with an unprecedented global spatial resolution despite variation within its measurement swath. Establishing robust global river elevation datasets for the pre-SWOT period is critical to prepare for the SWOT mission and for the study of hydrology more broadly.

Satellite radar altimetry data have enabled important scientific advances in hydrology (Birkett et al., 2002; Bjerklie et al., 2005; Calmant et al., 2008; Jung et al., 2010, Guetirana et al., 2009, Birkinshaw et al., 2014, Frappart et al., 2015, Becker et al., 2018, Emery et al., 2018, among many others), but spatial coverage is limited. This is for two primary reasons: inclination or latitude coverage limits of radar altimeter orbits (orbits with better temporal resolution have worse spatial coverage), and technical measurement challenges associated with retrieving elevation over seasonally varying rivers. Indeed, radar altimeter orbits and elevation retrieval technology were originally designed for characterizing ocean surface topography. The orbital characteristics of historic and contemporary radar altimetry missions used for hydrology tend to follow either the 10-day TOPEX/POSIEDON/Jason-1/-2/-3 orbit with relative high temporal resolution but low spatial coverage, or the 35-day ERS-1/-2/Envisat/SARAL-AltiKa orbits with low temporal resolution but higher spatial coverage. Neither of these orbit paradigms capture all global rivers (Alsdorf et al., 2007).

The second fundamental cause of poor global coverage of river radar altimeter observation availability is rooted in the measurement itself. There are a set of criteria, such as river width, nearby topography, and groundcover, associated with successful water surface level retrieval, but none have been shown to be fully predictive of water level accuracy (Maillard et al., 2015). Most of Earth's rivers are too narrow to accurately be accurately measured by satellite radar altimeters: Lettenmaier et al. (2015) suggest that rivers should be wider than 1,000 m for optimal retrieval, primarily due to the 1–2 km footprint size of pulse-limited satellite altimeters. Radar altimeter effective footprint size is a function of the surface characteristics and pulse emission mode. For example, in Low Resolution mode (LRM), which was commonly used for satellite altimeters until ~2016, footprints typically range from 1.5 to 6.0 km in diameter, depending on the land topography

near rivers. Thus, all but the widest rivers are (technically) sub-footprint features in LRM. Radar altimetry retrieval of river surface elevations thus relies on the fact that rivers reflect more radar signal than does land, due to the high dielectric constant of water. Some studies have developed methods to process radar altimetry data for far narrower rivers with LRM altimeters (e.g. ~100 m) for a particular location (e.g. Santo da Silva, 2010., Maillard et al., 2015, Boergens et al., 2016, Biancamaria et al.,2017). Since ~2016, retrieving water levels over narrow rivers is increasingly common with the Synthetic Aperture Radar (SAR) altimetry missions (e.g. Cryosat-2 and Sentinel-3) for which the equivalent footprint (300 m wide along flight track band) enables much easier detection and processing of radar returns from rivers.

Regardless of the specifics of a particular measurement location, altimeter range data (direct sensor measurement) requires a great deal of processing to be converted into usable surface heights. Measurements of ocean height rely on an onboard processor known as a "tracker" to dynamically estimate the approximate range of the target (i.e. the sea surface) in order to map received radar pulses to precise surface elevations. The onboard tracker works well for measuring ocean surface elevations, but it is unsuitable for estimating continental surface elevations. It thus requires further processing steps, known as "retracking". Using retracked river observations, inland radar altimetry can accurately measure changing river surface elevation (Koblinsky et al., 1993, Berry et al., 2005; Frappart et al., 2006; Alsdorf et al., 2007; Santos da Silva et al., 2010; Papa et al., 2010; Dubey et al., 2015, Tourian et al., 2016, Verron et al., 2018). While custom retrackers have been derived and tested in particular locations (Huang et al., 2018; Maillard et al., 2015; Sulistioadi et al., 2015) the ICE-1 retracker (Wingham et al., 1986) is arguably the best compromise between being consistently reliable and available for many altimeter missions (Biancamaria et al., 2017; Frappart et al., 2006; Santos da Silva et al., 2010). While available globally, the ICE-1 retracked data must be extracted over river targets, and carefully filtered, to make them useful to global hydrological modeling applications.

The four currently available radar altimeter datasets for rivers represent tremendous technical achievements: 1) Hydroweb (hydroweb.theia-land.fr); 2) Database for Hydrological Time Series over Inland Waters (DAHITI) (dahiti.dgfi.tum.de); 3)River&Lake Near Real Time (NRT) (https://web.archive.org/web/20180721182437/ http://tethys.eaprs.cse.dmu.ac.uk/RiverLake/shared/main); and 4) HydroSat (hydrosat.gis.uni-stuttgart.de/php/index.php). However, they are not optimized for the specific needs of global hydrologic modelers, who require global coverage, and enhanced ease of use (accessibility and metadata). Note that River&LakeNRT is no longer online but we compare against it for historical reasons (an archive link has been provided). Existing datasets have several characteristics that make them challenging to use for global hydrologic modeling. First, they tend to include dense coverage where altimeters perform well (e.g. over large, tropical rivers), or based on programmatic priorities of funding agencies. Hydroweb has 991 river VSs in South America alone, for example, primarily in the Amazon basin, while most include few Arctic rivers. One challenge of including Arctic rivers involves the complicating effect of river ice, which is widespread for much of the year. Three of the four datasets (Hydroweb being the exception), cannot be downloaded in bulk, but require repetitive clicking via web interface.

In this study, we determined what fraction of available altimeter data would be useful for global rivers using retracked data available from the official distribution of the instrument data (the geophysical data records (GDR)), unsupervised methods, and automatic data filtering processes. The result is the Global River Radar Altimetry Time Series (GRRATS), a global river altimetry dataset comprised of an opportunistic exploitation of VSs on the world's largest rivers specifically suited for the needs of global hydrological applications. GRRATS uses the VS as its fundamental organizational element. VSs are locations where ground tracks of exact repeat altimetry mission orbits cross rivers, enabling the development of a time series of water elevation observations. VSs can be thought of much in the same way as an in situ river gaging station, but are entirely derived from remote sensed measurements of river surface elevation. GRRATS is an "Earth Science Data Record" (ESDR) hosted at Physical Oceanography Distributed Active Archive Center (PO.DAAC) with a focus on conforming to data management and stewardship best practices (Wilkinson et al., 2016). GRRATS currently spans 2002 – 2016, and includes global ocean-draining rivers greater than 900 m in width: these collectively drain a total of >34% of global land area. GRRATS follows data management best practices as outlined by Wilkinson et al. (2016), and it includes extensive metadata. In developing GRRATS, our purpose is to create an accurate dataset, and also to create a better data product focused on ease of use.

## 2 Methods

There are four major steps in building GRRATS (Durand et al., 2016): 1) identification of potential VSs on global rivers; 2) extraction of altimeter observations from the Geophysical Data Records (GDRs); 3) filtering out noisy returns from the altimetry; and 4) performing either quantitative of qualitative evaluation. The philosophy and overview of GRRATS methods are reviewed here, whereas details of GRRATS production are more thoroughly described in the User Handbook (ftp://podaac-ftp.jpl.nasa.gov/allData/preswot_hydrology/L2/rivers/docs/).

### 2.1 Identification of potential VSs

We began by identifying potential VS for GDR extraction by identifying locations on global ocean-draining rivers where altimeter orbital ground tracks cross river locations greater than 900 m in width. We chose 900 m as our lower width limit as previous work has shown that VSs with widths >1 km present a higher probability of good performance (Birkett et al., 2002; Frappart et al., 2006; Kuo and Kao, 2011; Papa et al., 2012). This selection of rivers is spatially varied and large enough to provide a sensible constraint on global models. We used the intersection of the nominal altimeter ground tracks with the Global River Widths from Landsat (GRWL) dataset to identify such locations (Allen and Pavelsky, 2018).

## 2.2 GDR extraction

We extracted altimeter observations at the VS from the GDRs; this consisted of three steps. First we spatially joined Landsat imagery (selected from times of mean river discharge) compiled for the Global River Widths from Landsat (GRWL) river centerlines dataset (Allen & Pavelsky, 2015; Allen & Pavelsky, 2018) with satellite ground tracks to define the width extent of the mask used for the extraction of water elevations. Each mask was constructed using the width extent and upstream and downstream limits that were 2km perpendicular to the crossing location. We extracted all altimeter returns with centroids falling within each mask for each pass from Jason-2 GDR version D (Dumont et al., 2009), and the Envisat GDR, Version 2.1 or later (Soussi and Féménias, 2009), using corrections outlined in product documentation. We extracted ICE-1 retracked ranges from the GDR (Gommenginger et al., 2011; Wingham et al., 1986). To get ellipsoidal heights, we applied the standard combination of parameters and corrections. We then converted these ellipsoidal heights to an orthometric height above the geoid, using the Earth Gravitational Model 2008 (EGM08) model (Pavlis et al., 2012).

## 2.3 Data filtering

We filtered altimetry data in a six-step process. First, we filtered using an a priori Digital Elevation Model (DEM) data baseline elevation (median of all best available DEM values falling within the extraction polygon) at each VS. We used Shuttle Radar Topography Mission (SRTM), Global Multi-Resolution Terrain Elevation Data (GMTED), and Advanced Spaceborne Thermal Emission and Reflection Radiometer (ASTER), in that order of preference(Abrams, 2000; Danielson and Gesch, 2011; Van Zyl, 2001). We filtered out elevations 15 m above or 10 m below the constrained baseline elevation. We arrived at these limits by examining over 150 United States Geological Survey (USGS) gages with upstream drainage areas larger than 20,000 km$^2$ and changing the upper filter limit (responsible for 90.5% of data points filtered due to height), to 14 m or 16 m resulting in a 4.2% increase and 3.8% decrease in filtered points respectively. We determined these limits should reasonably encompasses any measurements of the river surface as the Amazon flood wave is capped around 15 m from trough to peak (Trigg et al., 2009). Second, we applied an additional elevation filter removing any elevations that fell 2 m or more below the 5$^{th}$ percentile of surface elevations in the time series (0.03% of total returns). We obtained low-end filter criteria for removing observations impacted by near-river topography at low flow by trial-and-error. Third, we flagged and remove elevations from times of likely ice cover. Ice cover dates were determined from USGS and Environment and Climate Change Canada (ECCC) data when available. If ice breakup data were not available, we applied broad date limits regionally, using observations from the *Pavelsky and Smith* (2004) study of Arctic river ice breakup timing. Breakup dates range from late September to early June. Fourth, remaining elevations were averaged for each cycle at each VS. Fifth, we removed any potential VS with < 25% or 50% of available cycles for rivers with and without ice cover respectively. Finally, we determined a flow distance limit for tidal VSs (those where the tidal signal was dominant) using visual inspection of the time series on each river and removed VSs below that point.

## 2.4 Data Evaluation

We acquired evaluation stage data from 65 stream gages (on 12 rivers) (Environment Canada, 2016; Jacobs, 2002; Martinez, 2003; USGS, 2016). All stage data is publicly available with the exception of data from the Congo, Ganges, Brahmaputra, and Zambezi which was provided by the authors. Note that VSs rarely fall in the same location as a stream gage; thus, most
5 studies recommend some VS-in-situ stream gage distance (e.g. 200 km) beyond which comparisons are not performed (Michailovsky et al., 2012). Analyses showed that VS-stream gage distance was often not an accurate predictor of height anomaly differences. This is likely due to the hydraulics (width, nearby dams, confluences) of a more distant gage being more similar to the location of the VS than the most proximal gage. Thus, in this study, we compared each virtual station with all in-situ gages available on the main channel of that river. At each VS, we reported error metrics for the best, median,
and the spatially closest comparison. For completeness, we included VSs with poor error metrics; users can then select which of the VSs to use, based on their reported error statistics and the user applications. Following the normal practice in the field (e.g. Berry, 2010; Schwatke et al., 2015), we compare relative heights between VSs and gages, as opposed to absolute heights, in order to avoid the influence of difference in datum and the lack of correspondence between satellite ground tracks and gage locations. We calculated relative heights by removing the long-term mean between the sample pairs of VS heights
and the stage measured by the stream gages. Error metrics in GRRATS include the correlation coefficient (R), Nash-Sutcliffe Efficiency (NSE), and Standard deviation of the errors (STDE). NSE is typically employed to describe the goodness of fit for a modeled result with measured values, so our use here is non-traditional. Nonetheless, we use NSE because, as opposed to R and STDE, NSE normalizes error with variation from the mean in the observed, or in our case, in-situ data, by comparing error to actual variability. For example, 1 m of error can be an issue of varying severity when rivers
can have height variation ranging from >10 m (Amazon) to <5 m (St Lawrence). It is also an established metric for goodness of fit within the altimeter literature (Biancamaria et al., 2018; Tourian et al., 2016).

While qualitative grades are not as reproducible as best fit statistics, they have been used in the past to guide users to preferable time series when no other error metrics are available (Birkett et al., 2002). For the remainder of our VSs (without stage gages), we performed a qualitative evaluation of the station represented by a letter grade ranging from A (highest level
of confidence on the data quality) to D (lowest level of confidence). The criteria used in the assignment of letter grades was based on the presence of obvious outliers, number of data points in the time series, and time series continuity with nearby VSs. We determined outliers by visual inspection. Letter grades are take in to consideration all of these criteria, but in general, VSs with an A rating would have 1 or fewer obvious outliers per year, no more than 2 cycles filtered out per year, and will fit nicely above VS downriver and below VS upriver. A D rating might be applied to a VS with 3 or more outliers
30 per year, 5 or more cycles missing per year, and might fall below VS downriver from it, and above VS upriver from it. We explicitly recorded and document which VSs in GRRATS are evaluated using this qualitative approach.

## 3. Results and Discussion

GRRATS processing produced a total of 932 globally distributed virtual stations (Figure 1). The 39 GRRATS rivers account for 50M km² (>34%) of global drainage area, and include 13 Arctic rivers. To attain these results, we extracted and processed a total of 1.5M individual radar returns at 1478 potential VS locations.

### 3.1 Filtering returns

We removed 309.7K altimetry returns with our height filters (steps 1 and 2 of our filtering process), leaving 1.1M (78.2%) viable measurements. Our ice filter removed an additional 296.9K of the remaining returns (step 3) resulting in 810.4K viable returns (57.2%). Averaging all height returns within the river polygons for each pass at each VS (step 4) led to a total of 102.3K (21.9K on Arctic rivers) pass-averaged measurements. VSs were required to retain 50% (without ice) or 25% (with ice) of their passes post-filtering to be included in the final data product, resulting in the removal of 465 potential VS locations (step 5). VSs were also removed by visual inspection if they were tidal, resulting in the removal of an additional 45 stations (step 6). While many VSs were filtered heavily, 72.8% of the total returns for all VSs in the final product passed all filters (the median VS value being 97.7%) and 227 VSs lost no returns. The filtering process resulted in a total 932 VSs for evaluation derived from standard retracked data (ICE 1). These VSs had a data set wide average of ~16 measurements per year (9.5 for Envisat VSs and 35.8 for Jason2 VSs).

### 3.2 Example Time Series evaluation

Figure 2 shows example GRRATS time series for the Mackenzie and Amazon Rivers and corresponding in-situ gages. Error bars represent the range of the values that were averaged to generate each data point (does not include filtered data points). Data necessary to compute error bars are a part of the data product. Comparison between the Jason-2 time series and the gage on the Mackenzie River produced STDE = 0.5 m, NSE = 0.41, and an R = 0.64. In this case, the gage used for evaluation was located ~700 km upriver (Figure 2(a)). The STDE is approximately consistent with what is expected from the literature (Asadzadeh Jarihani et al., 2013; Frappart et al., 2006). However, the STDE is relatively large in comparison with the overall annual range in the time series (typically ~2 m) observed from the gage (see Figure 2 (a)), leading to a relatively low NSE. Additionally, several cycles have far larger errors, reaching up to two meters, in some cases. There are a total of 3 in-situ gages used for evaluation on the Mackenzie River. Across the 3 gage comparisons, this VS had median statistics of 0.58 m, 0.35 and 0.64 for STDE, NSE, and R, respectively. Comparing the VS data to the gage on the Amazon River yields STDE= 0.98 m, NSE= 0.94 and R= 0.97, with the evaluation gage 263 km upriver from the VS (Figure 2(b)). Despite the STDE being nearly twice as large, the magnitude of change on the Amazon allowed for a much better fit due to the large interannual variability of the Amazon floodwave (>10 m). Most of the error was from times of low flow near the ends of the calendar year in 2009, 2011 and 2012. There are 6 in-situ gages on the Amazon River. Across these comparisons, this VS had median statistics of 0.94 m, 0.95, and 0.98 for STDE, NSE, and R, respectively.

### 3.3 GRRATS evaluation across all rivers

We compared GRRATS against in-situ evaluation data on a total of 12 rivers. This provided evaluation of 380 of the 920 virtual stations (42%). On each river, the total number of time series evaluations was the product of the number of VSs and the number of gages (Figure 1). Thus, the total number of time series evaluations (summed across all 12 rivers) was 1,915 (Table 1).

A total of 72.5% of the quantitatively evaluated virtual stations had an NSE greater than 0.4 when compared with at least one gage. The highest maximum NSE (Figure 3(a)) was 0.98, from an Envisat VS in the upper reaches of the Amazon. The median value for maximum NSEs for all VSs was 0.75 (0.67 from closet gage comparison Figure 3(c)). A total of 341 of the 389 (87.7 %) virtual stations had a maximum NSE >0 (Figure 3(a)) .The highest median NSE (Figure 3(b) and values were 0.96 at two Envisat VS on the Orinoco river (lower and mid). A total of 277 of 389 (71.2%) had a median NSE >0.

The smallest minimum STDE (to two significant digits) was 0.11 m and occurred at an Envisat VS on the upper Congo. The median value for minimum STDE (Figure 3(d)) for all VSs was 0.93 m (1.08 m from closest gage comparison Figure 3(f)). The minimum and median value for median STDE (Figure 3(e)) were 0.31 m, and 1.3 m respectively. Our STDE error statistics are greater than previous work reporting accuracies ranging from 0.14 m to 0.43 m for Envisat data and 0.19 m to 0.31 m for Jason-2 data (Frappart et al., 2006; Kuo and Kao, 2011; Papa et al., 2012; Santos da Silva et al., 2010). This discrepancy is likely because GRRATS includes VSs on rivers where evaluations have not previously been reported in the literature, and the fact that we do not fine-tune processing or filtering to each VS due to the global nature of the dataset.

Some locations with relatively low STDE values showed poor performance in terms of NSE, particularly for rivers with relatively low water elevation variability. VSs on the St Lawrence River had minimum STDE ranging from 0.58 - 3.27 m. The VS with 0.58 m STDE corresponded with a maximum NSE value of -0.27, indicating quite poor performance in resolving river variations (standard deviation of 0.35 m). The St Lawrence River is anomalous in other ways as well. For 2 potential VSs (one each from Jason 2 and Envisat), the unprocessed data (ICE-1 retracked GDR data) showed a bias of several tens of meters above the baseline height, and thus no data for these VSs are included in GRRATS. Closer examination of these VSs seems to indicate that the on-board tracking window was often tens of meters outside of the river surface range, making retrievals from the surface impossible. This case is particularly odd as such errors are not expected for wider rivers; the St Lawrence is between 2 and 7 km wide where we sampled it. Such errors are more commonly associated with altimeter returns from near-river topography on narrow rivers (Biancamaria et al., 2017; Frappart et al., 2006; Maillard et al., 2015; Santos da Silva et al., 2010). Moderately poor performance from the remainder of VSs in terms of NSE and STDE on the river is likely due to the river lacking enough variation in height to allow for retrieval of a good signal outside the error range of radar altimeters. However, this low variation data can still be quite useful to modelers for determining if their results show excessive change in the annual cycle of water elevations.

The median of the maximum R values (Figure 3(g)) for each station is 0.9 (0.87 from closest gage comparison Figure 3(i)). The maximum R value plot shows left skewness, similar to the NSE results. The lowest maximum R value of -0.15 occurred

at an Envisat VS on the mid St Lawrence River, which was the only virtual station to display a negative correlation. The best maximum R value was 0.99 for an Envisat station near the mouth of the Ganges River that also displayed high NSE and low STDE. The median value of the median R (Figure 3(h)) is 0.69. The values range from -0.18 (an Envisat VS on the lower St Lawrence) to 0.99 (an Envisat VS on the lower Brahmaputra).

For 27 of the 39 rivers in the GRRATS dataset, no in-situ data is available for evaluation. We gave the remaining 27 rivers qualitative letter grades based on number of missing data points, obvious outliers, and agreement with nearby stations. These grades are included with the data for end users (Table 2). The majority of rivers evaluated this way fall into the B or C category (~61%), with only ~15% getting an A rating.

## 3.4 Towards quantitative performance prediction

As is evident above, radar altimeter performance varies dramatically across rivers and across VSs. Generally, measurements from wide rivers without large topographic features in the altimeter footprints that have large seasonal water elevation variations tend to result in better altimeter performance. In order to identify conditions that may contribute to poor return quality, we compared both VSs width and percentage of original returns post-filtering, near-river topography, and river height variation with all three fit statistics. We found no statistically significant relationships in this evaluation, a finding that

supports existing literature on quantitative prediction of altimeter performance (Maillard et al., 2015). Indeed, we found many examples of counterintuitive performance in our examination. The St. Lawrence (described above) is an example of unexpectedly poor performance; typical predictors such as width (smallest VS ~1.5 km wide) and the lack of extreme proximal topography led to an expectation of accurate performance that was not met. Meanwhile, other rivers defied the normal pattern by showing good fit metrics while being far narrower. The Mississippi River was consistently at our lower

limit for river width. The VS widths ranged from 509.1 m to 2,608.0 m, and had an average width of just 955.3 m. The average near-river relief ranged from 10-60 m. The Mississippi maximum NSE values ranged from -0.22 to 0.96, with an average of 0.43. Minimum STDE values ranged from 0.34 m-2.22 m, with an average of 1.18 m. additionally, we computed average error statistics across all VSs along each river. Some rivers stood out as particularly good or poor performers (Table 3), but no broad geographical patterns emerged. For this reason, we recommend using the median (dataset wide) value for

evaluated STDE (0.93 m) as an error estimate for VSs without evaluation data, as this is representative of 42% of all of the VSs in the dataset. While we do not provide error estimates at the individual data point level, we suggest that individual VS data point error be treated as the STDE of the time series they are a component of.

## 3.5 Comparison to other altimetry datasets

While it is outside the scope of this study to compare GRRATS exhaustively with existing datasets, we find it appropriate to

demonstrate that our dataset is comparable. Therefore, we compared three VS locations that are in each of the four datasets discussed (one on the Amazon, Congo, and Brahmaputra). Figure 4 (a-c) show time series anomaly at each VS and the closest gage. Note that time series lengths are limited to the shortest time series in the comparison and do not match the

coverage of any particular mission, and that River&LakeNRT data was unavailable for the VS location on the Brahmaputra GRRATS, DAHITI, and Hydroweb are similar and fit with the in-situ gage well (Table 4). DAHITI is missing data on the Amazon time series. HydroSat and River&LakeNRT are frequently out of phase, particularly on the Amazon River (Figure 4(a)). Performance is similar on ungaged rivers when compared (Figure 5). GRRATS and DAHITI showed good agreement

on the Parana River (Figure 5(a)). HydroSat and Hydroweb (Figure 5(b-c)), are differentiated from GRRATS on the Ob' and Lena Rivers, as they show heights from a frozen river that GRRATS flags and removes. During overlap, HydroSat and GRRATS were similar at the Ob' VS. Hydroweb data on the Lena is similar to GRRATS, with the exception of the 2006 peak flow, which is missing. Note that much of the rising limb is missing in these time series as it occurs during times of ice cover. Unfiltered data and ice flags are available to data users if needed. This process demonstrated that our quasi-automated

methods produce a dataset with global coverage and performance that approximates the accuracy of regional altimetry datasets.

## 4. Data availability

GRRATS (DOI 10.5067/PSGRA-SA2V1) is available at

https://podaac.jpl.nasa.gov/dataset/PRESWOT_HYDRO_GRRATS_L2_VIRTUAL_STATION_HEIGHTS_V1for non-

commercial use only (Durand et al., 2016). Data are provided in NETCDF format. For a file content description please see Appendix A. An interactive map of the data is located at http://research.bpcrc.osu.edu/grrats/. This tool is intended for exploration only, and may not reflect the most up-to-date version of the data. As with figure 2, error bars represent the range of the values that were averaged to generate each data point (does not include filtered data points). Data necessary to compute error bars are a part of the data product.

## 5. Conclusion

We find that uniform altimeter data processing produces usable data with accessible documentation for end users. Encouraging end user understanding of how this kind of data is produced is critical in fostering its use across the scientific and stakeholder communities. GRRATS considers only ocean-draining (highest order) rivers, while other datasets include some VSs on large tributaries. However, our use of the GRWL dataset allowed for a comprehensive selection of altimeter

crossings on a global scale. These features should enable broad use by the scientific community. This resulted in GRRATS having the best coverage available for North American rivers as well. We produced GRRATS with ease of use in mind. VS metadata are included and the product can be downloaded in bulk.

On the whole, the median value of the error standard deviation is 0.93 m, which is similar to or slightly larger than values reported for the rivers that are most commonly studied using radar altimetry (e.g., the Amazon and Congo). Our philosophy

in constructing the dataset was to maximize the spatial coverage of altimeter crossings, construct the product in a uniform

way, and to provide an evaluation of quality for each VS. Thus, users can decide whether each VS is useful given their data needs. Note that a total of 77.2% of virtual stations evaluated against in-situ data had an NSE>0.4. Our uniform production method allowed us to evaluate whether river width or the height of bluffs proximal to rivers at altimeter crossings correlate with altimeter performance, as was expected in the literature. However, we were unable to identify a predictive model for altimeter performance, and leave this exercise for future work.

The GRRATS dataset maximizes traceability: all of the information needed to re-process these VSs is included in the final data product. It is our expectation that other researchers could implement other methods of filtering and processing to achieve derived data products tailored to their applications.

## Author Contributions

SC1 developed and finalized processing algorithms, performed methods exploration, was primary data manager, analyzed final product, and finalized manuscript. MD developed algorithms, performed quality analysis, and provided editorial and graphical assistance. YY performed GDR extraction and geodetic corrections and was primary unprocessed data manager. YJ performed methods exploration. QG performed methods exploration. ST developed algorithms and performed methods exploration. CKS provided technical expertise and editorial assistance. GHA and TP provided access to GRWL width data used to find virtual station targets. SC2 provided technical expertise, in situ data, and editorial assistance.

## Acknowledgments

This work was produced on behalf of the NASA MEaSUREs group (grants NNX13AK45A, and NNX15AH05A). We would like to acknowledge the use of stream gage data and imagery obtained from the USGS stream gage network and other organizations, and USGS Landsat archive, respectively; as well as altimeter VS data products from other research institutes.

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

**Tables**

**Table 1 Summary Statistics from section 3.3**

| Fit Statistic | Best | | Closest | Median | |
|---|---|---|---|---|---|
| NSE | Highest | Median | Median | Highest | Median |
| | 0.98 | 0.75 | 0.67 | 0.96 | 0.31 |
| R | Highest | Median | Median | Highest | Median |
| | 0.99 | 0.9 | 0.87 | 0.99 | 0.69 |
| STDE | Lowest | Median | Median | Lowest | Median |
| | 0.11 m | 0.93 m | 1.08 m | 0.31 m | 1.3 m |

**Table 2 Qualitative letter grade summary**

| Grade | A | B | C | D |
|---|---|---|---|---|
| Number of VS with grade | 85 | 155 | 177 | 114 |

**Table 3 River Average fit statistics**

| | Best Average Statistics | | Worst Average Statistics | |
|---|---|---|---|---|
| Fit Statistic | River | Value | River | Value |
| Maximum NSE | Brahmaputra | 0.82 | St Lawrence | Max NSE<0 |
| | Orinoco | 0.78 | Susquehanna | |
| | Amazon | 0.69 | Columbia | |
| | Ganges | 0.65 | Mackenzie | |
| | Congo | 0.6 | | |
| Maximum R | Orinoco | 0.93 | St Lawrence | 0.3 |
| | Brahmaputra | 0.92 | Mackenzie | 0.46 |
| | Ganges | 0.87 | Columbia | 0.49 |
| | Congo | 0.85 | Susquehanna | 0.68 |
| Minimum STDE | Congo | 0.53 m | Mekong Orinoco | 2.61 m |
| | Yukon | 0.76 m | | 1.95 m |
| | Brahmaputra | 1.07 m | Mackenzie | 1.88 m |
| | Mississippi | 1.18 m | St Lawrence | 1.69 m |

**Table 4 Multi-product fit statistics from figure 5**

| Product | STDE | R | NSE |
|---|---|---|---|
| Amazon River | | | |
| HydroSat | 2.12 m | 0.61 | 0.33 |
| Hydroweb | 1.42 m | 0.96 | 0.72 |
| NRTRL | 2.9 m | 0.3 | -0.74 |
| DAHITI | 0.85 m | 0.99 | 0.81 |
| GRRATS | 1.57 m | 0.95 | 0.65 |
| Congo River | | | |
| HydroSat | 0.48 m | 0.87 | 0.76 |
| Hydroweb | 0.42 m | 0.92 | 0.84 |
| NRTRL | 3.2 m | 0.11 | -7.88 |
| DAHITI | 0.39 m | 0.93 | 0.86 |
| GRRATS | 0.5 m | 0.91 | 0.81 |
| Brahmaputra River | | | |
| HydroSat | 0.56 m | 0.96 | 0.92 |
| Hydroweb | 0.58 m | 0.91 | 0.96 |
| DAHITI | 0.6 m | 0.96 | 0.86 |
| GRRATS | 0.69 m | 0.95 | 0.87 |

**Figures**

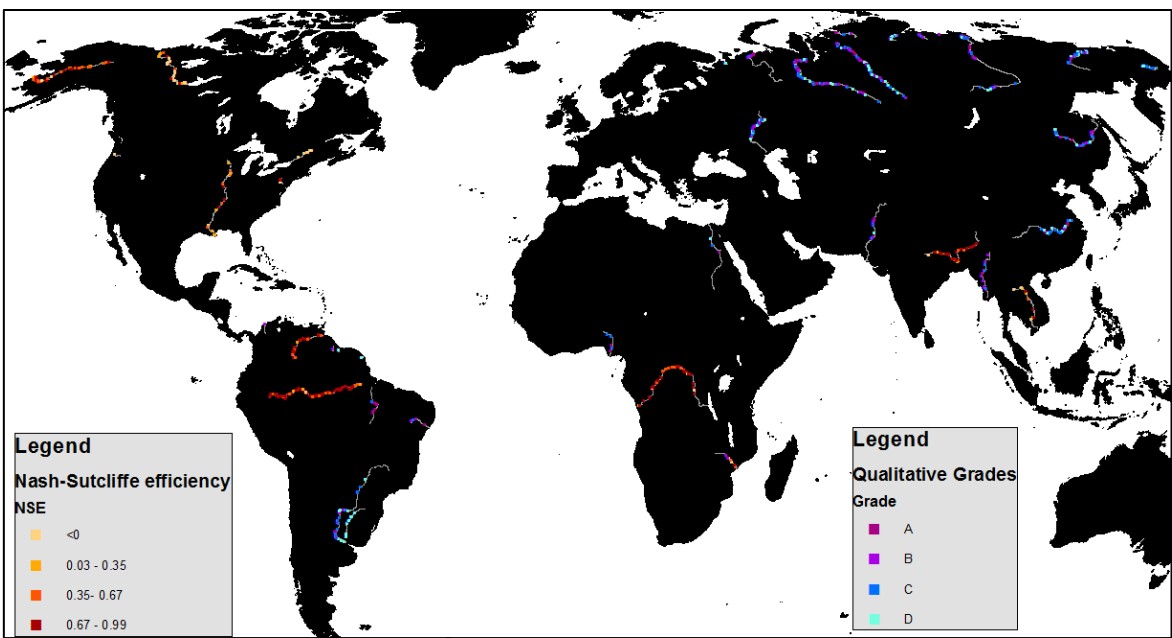

**Figure 1 The GRRATS dataset and evaluation results. Maximum NSE (best fit) plotted in yellow to red (shown on all rivers with gage data) and qualitative grades plotted in teal to dark purple. In both cases, darker colors indicate better evaluation results. Each river is evaluated using only one of these methods.**

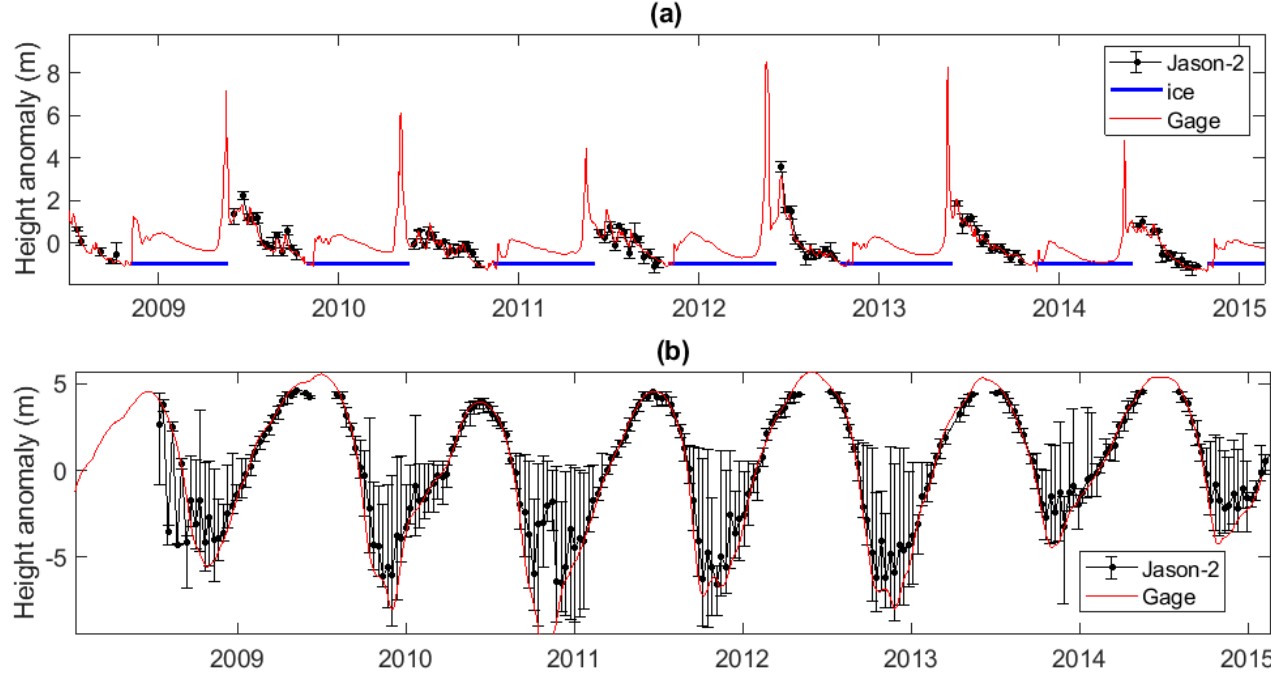

**Figure 2 Example time series for the Mackenzie River. Panel (a) shows water surface heights with ice filtering compared to Environment Canada gage (10KA001) located 684 km away from the virtual station. Panel (b) compares the time series derived from Jason-2 for one of the Amazon gages. Error bars represent the range of the values that were averaged to generate each data point (does not include filtered data points).**

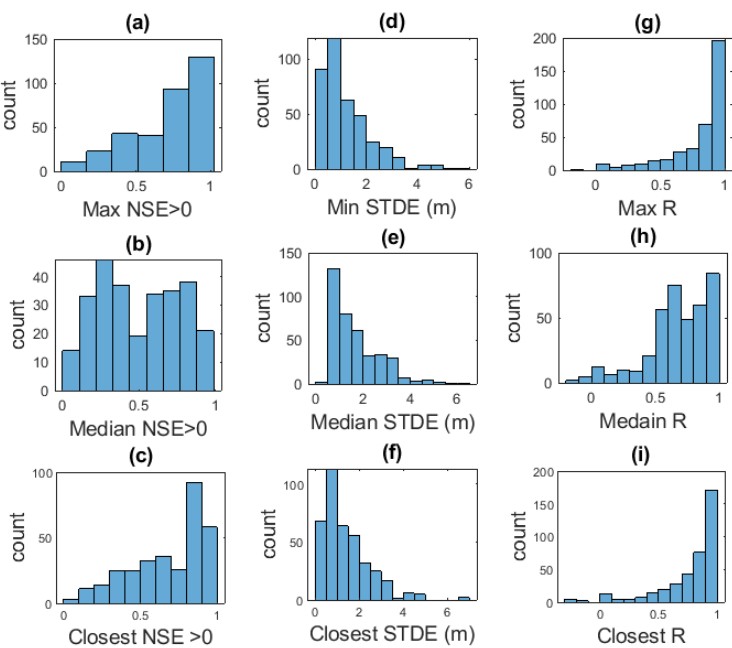

**Figure 3 Virtual Station fit statistics computed with all available evaluation gages located in the same river and closest comparison. Please note that NSE values are plotted here only when greater than 0 to enable readers to more easily see the majority of the data. 12.2%, 28.8% and 17.2% of the total data are not shown in panel a, b, and c respectively. Panel (a) histogram of the max NSE >0 at each VS in the dataset, Panel (b) histogram of the median NSE > 0 at each VS in the dataset, Panel (c) histogram of closest NSE>0 , Panel (d) A histogram of the minimum STDE in the dataset, Panel (e) A histogram of the median STDE all theVSs in the dataset,Panel (f) histogram of closest STDE, Panel (g) histogram of the max  R at each VS in the dataset, Panel (h) histogram of the median R at each VS in the dataset, Panel (i) histogram of closest R.**

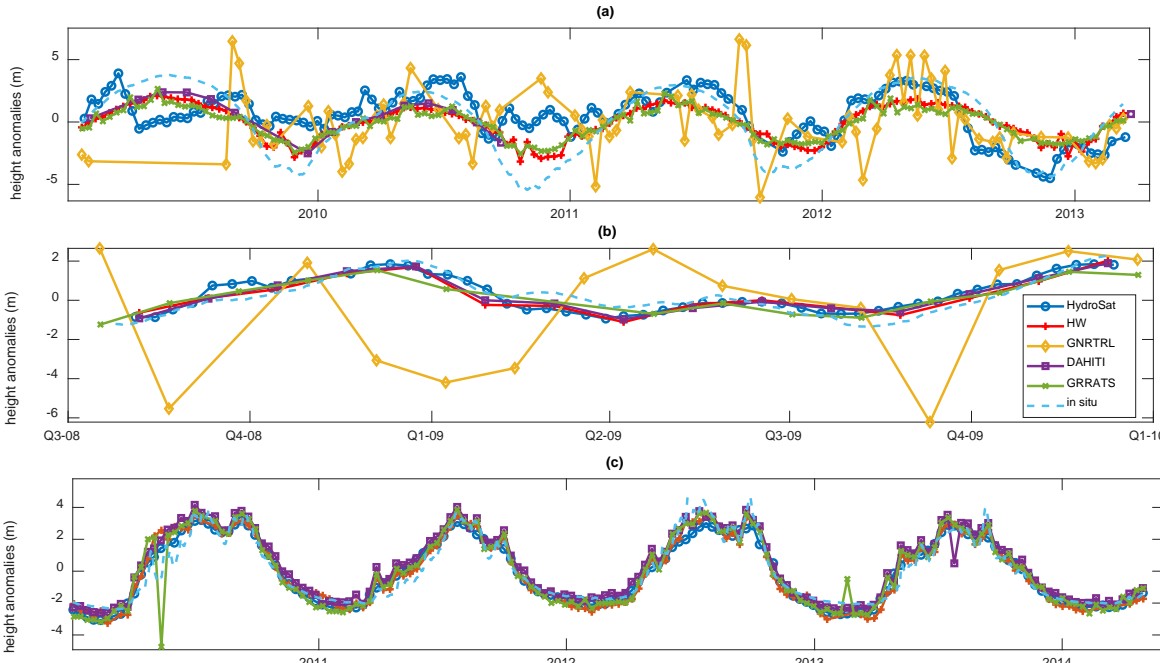

**Figure 4: Multi product evaluation at same location.  Panel (a): multiproduct comparison on the Amazon River Panel (b): multi product comparison on the Congo River. Panel (c): multi product comparison on Brahmaputra River. DAHITI plotted in purple with square markers, HydroSat in dark blue with circle markers, River&LakeNRT in yellow with diamond markers, Hydroweb in red with cross markers, and GRRATS in green with x markers and in-situ in dashed light blue. Note that the legend in panel (b) apples to all of figure 4. GRRATS error bars not shown to improve readability.**

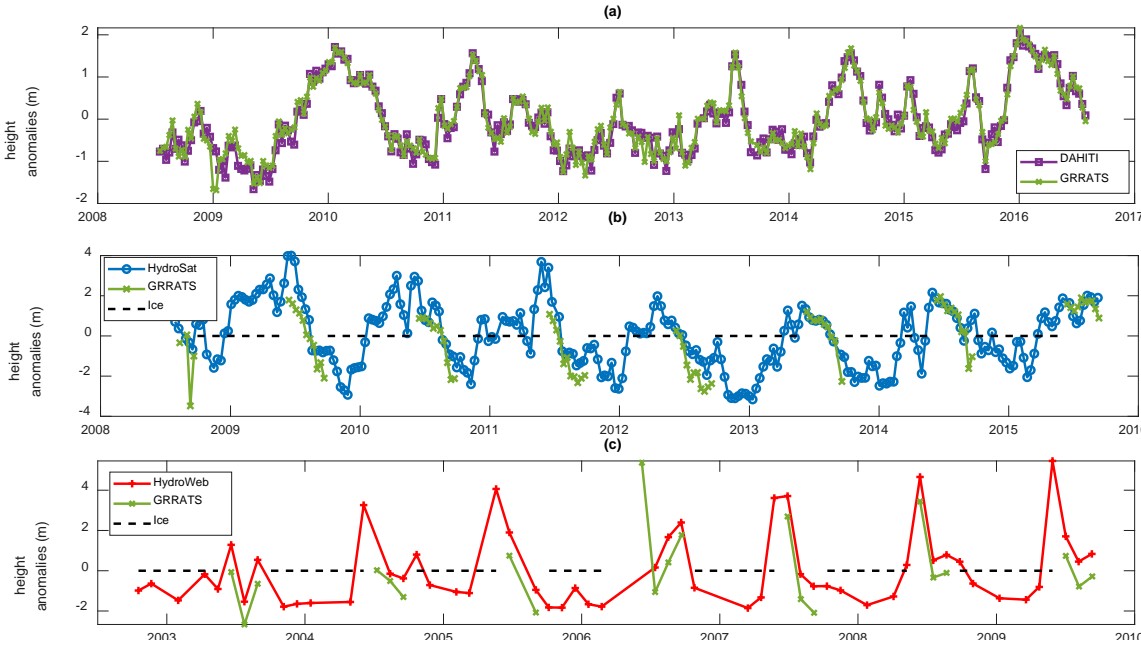

**Figure 5 multi product evaluation at ungagged river locations GRRATS plotted in green, DAHITI in purple with square markers, HydroSat in blue with circle markers, HydroWeb in red with cross markers, and times of ice cover plotted with a dotted black line. Panel (a) is a comparison with DAHITI on the Parana River. Panel (b) is a comparison with HydroSat on the Ob River, and Panel (c) is a comparison with HydroWeb on the Lena River.**

**Appendix 1**

## Data packaging and Variable identification

**Sample Altimetry Data (NetCDF format)**

Format: netcdf4                                    Title = 'Altimetry Data for virtual station Yukon_Jason2_0'

<p align="center">Global Variables:</p>

| Variable | Dimension | Datatype | Units | Name |
|---|---|---|---|---|
| lon | X | double | degrees east | longitude |
| lat | Y | double | degrees north | latitude |
| ID | root | char | - | Reference VS ID |
| sat | root | char | - | satellite |
| Flow_Dist | distance | double | km | Distance from river mouth |
| rate | root | double | Hz | sampling rate |
| pass | root | int32 | - | pass number |
| nse | grade | double | - | Max Nash Sutcliffe efficiency |
| nse AVG | grade | double | - | Average Nash Sutcliffe efficiency |
| R | grade | double | - | Correlation coefficient |
| std | grade | double | m | Minimum standard deviation of error |
| stdAVG | grade | double | m | Average standard deviation of error |
| grade | grade | char | - | qualitative letter grade |

The global variables are: longitude and latitude of the center of the virtual station, the virtual station ID, the satellite name, flow distance, sampling rate , the satellite pass number and a suite of fit statistics, or a qualitative letter grade. Qualitative letter grades were assigned based on amount of data points, seasonal pattern, and similarity to nearby VS. This was done, only when validation data was unavailable. When validation was possible, the VS was evaluated with all gauges on the river through relative height comparison. Maximum Nash-Sutcliffe Efficiency (NSE), Average NSE, maximum R(correlation coefficient), minimum standard deviation of error (STDE), and average STDE are reported.

<p align="center">Groups:</p>

| /Unprocessed GDR Data[/ | | | | |
|---|---|---|---|---|
| Variable | Dimension | Datatype | Units | Name |
| lon | X | double | degrees east | Longitude |
| lat | Y | double | degrees north | latitude |
| h | Z | double | meters above EGM2008 geoid | Unprocessed heights |
| sig0 | UGDR | double | dB | Sigma0 |
| pk | UGDR | double | unknown | peakiness |
| cycle | UGDR | int32 | unknown | Altimeter cycle |
| time | T | double | Days since Jan-1-1900 00:00:00 | |
| heightfilter | UGDR | int32 | -flag- | Good heights flag |
| icefilter | UGDR | int32 | -flag- | No ice flag |
| allfilter | UGDR | int32 | -flag- | Ice free heights that passed height filter |

This includes the data from each return: lon and lat, the height of the water level in meters, the signal strength, sigma0, in decibels, a 'peakiness' value, the cycle number, the time of the return, and filter flags that signal 1 for data that should be included and 0 for data that should be excluded. The flags are for a height filter, an ice filter, and the logical intersection of the two
5 (allfilter), with 1 denoting returns that pass through the filter and 0 denoting returns that do not.

| /Timeseries/ | | | | |
|---|---|---|---|---|
| Variable | Dimension | Datatype | Units | Name |
| time | T | double | Days since Jan-1-1900 00:00:00 | time |
| cycle | TS | int32 | - | Altimeter cycle |
| hbar | Z | double | meters above EGM2008 geoid | average height |
| hwbar | Z | double | meters above EGM2008 geoid | weighted average height |
| sig0bar | time | double | dB | average sigma0 |
| pkbar | time | double | - | Average peakiness |

These are pass-averaged values, having gone through the filter. There are two values that flag data: -9999 for data that is missing from the GDR, and -9998 for data that is missing because of height/ice filters. These flags are only present when none of the
10 values to be averaged can be found. The other values give average height (hbar), in meters, and sigma-0 weighted height using

| /Sampling/ | | | | |
|------------|-----------|----------|----------------|----------------------|
| Variable | Dimension | Datatype | Units | Name |
| scene | scene | char | - | Landsat Scene ID |
| lonbox | X | double | degrees east | Longitude box extent |
| latbox | Y | double | degrees north | Latitude box extent |
| island | scene | int32 | -flag- | Island flag |

This is the data from the polygons, including the Landsat scene ID used to draw the polygons. The island flag is used when islands are visible inside the polygon in the imagery when drawing the mask.

| /Filter/ | | | | |
|----------|-----------|----------|---------------------------------|-------------------------------|
| Variable | Dimension | Datatype | Units | Name |
| nNODATA | - | int32 | count | Number of cycles without data |
| riverh | Z | double | meters above EGM2008 geoid | River elevation from filter file |
| maxh | Z | double | meters above EGM2008 geoid | Max elevation allowed by filter |
| minh | Z | double | meters above EGM2008 geoid | Min elevation allowed by filter |
| icethaw | T | double | Days since Jan-1-1900 00:00:00 | Thaw dates for river |
| icefreeze | T | double | Days since Jan-1-1900 00:00:00 | Freeze dates for river |
| DEMused | DEM | Char | - | DEM used in height filter |

This is the filter data; nNODATA gives the number of cycles that have no data because of a lack of data in the GDR and/or data that is filtered out. riverh gives the river elevation extracted from a 30 arc-second DEM of the region. This is used for the height filter. maxh and minh are the upper and lower bounds of river heights included in the filtered data; we set a +15m, -10m from the DEM river elevation as a first pass, and then removed any data that was 5m below the 5th percentile of river stage heights.
10  icethaw and icefreeze are the thaw and freeze dates, respectively, for the years included in the altimetry dataset. DEM used refers to the DEM that the bassline height was taken from.