# Peer review of "Global River Radar Altimetry Time Series (GRRATS): New River Elevation Earth Science Data Records for the Hydrologic Community"

_Earth System Science Data, 2019_

## Referee Comment (RC1) · Anonymous Referee #1 · 19 Aug 2019

In this paper the authors describe a fourteen year dataset of global radar altimeter-derived stream elevations. It is a good paper describing a valuable dataset. It is a definitely unique. My comments generally have to do with presentation, but there are two major problems. These comments are meant to improve readability of the paper and make a few specific concepts more clear. I hope my comments are constructive.

Most importantly, when I tried to access the data via the link provided in Section 4, I could not. Please verify the url is correct. I think the 'ftp' should be 'http'. I was able to get to the data viewer site at OSU, but this only provides images, and not the data itself. I was able to perhaps get to the correct data site. However, it seemed the data there

were metadata and not the time series of river elevations. Since I could not access the data I cannot provide a thorough review of the data usefulness or completeness. I apologize if I am missing something.

Second, the article needs more information, or a description, of the concept of a virtual station. It is never really clearly described what a VS is and thus it may be difficult for someone new to the field and data to understand what is being compared to in situ data.

Some specific comments include: 1) Page 1 Line 20: in "either quantitatively for VS where . . ." should it not be "VS levels where . . .." 2) Page 1 Line 21: As these are multiple VS, should this be VSs? I also wonder about the grammatical use of VS throughout the paper. 3) Page 3 Line 27: This is the first instance of VS in the main text, please spell it out. 4) Page 3 Line 28: Same with GRRATS, and with all the several acronyms throughout the paper 5) Page 4 Line 29: Its unclear where this polygon came from. Would calling it the 'mask' be more informative, as it was earlier? How was this polygon defined? This is not described in the text. 6) Page 6 Line 20: Maybe rename this section 'Data Description' 7) Page 6 Line 21: is this a typo: '50.M'? 8) Page 9 Line 16: The authors note there are two streams illustrated in Figure 4, but there are three. 9) Figure 1: Is the black background really the best choice? 10) Figure 4: A legend would be very helpful. Same in Figure 5.

---

## Short Comment (SC1) · 19 Aug 2019

We thank the reviewer for thier comments and wish to provide an updated link to the data as access has changed since submission. We will respond to the remainder of the reviewers comments shortly.

https://podaac.jpl.nasa.gov/dataset/PRESWOT_HYDRO_GRRATS_L2_VIRTUAL_STATION_HEIGHTS_V1

---

## Author Comment (AC1) · 21 Aug 2019

We would like to thank the reviewer for the time and effort they have put into reviewing our manuscript, as well as for the helpful comments they have provided. We have copied the reviewer's comments in black, and added our responses in red type (in the attached pdf version as this text field is plain text). Comment 1: "Most importantly, when I tried to access the data via the link provided in Section 4, I could not. Please verify the url is correct. I think the 'ftp' should be 'http'. I was able to get to the data viewer site at OSU, but this only provides images, and not the data itself. I was able to perhaps get to the correct data site. However, it seemed the data there were metadata

and not the time series of river elevations. Since I could not access the data I cannot provide a thorough review of the data usefulness or completeness. I apologize if I am missing something."

Response: The link to access the data has changes since the original submission of this paper. We have updated section 4 to read as follows:

GRRATS (DOI 10.5067/PSGRA-SA2V1) is available at https://podaac.jpl.nasa.gov/dataset/PRESWOT_HYDRO_GRRATS_L2_VIRTUAL_STATION_HEIGHTS_V1for non-commercial use only (Durand et al., 2016). An interactive map of the data is located at http://research.bpcrc.osu.edu/grrats/. Comment 2: "Second, the article needs more information, or a description, of the concept of a virtual station. It is never really clearly described what a VS is and thus it may be difficult for someone new to the field and data to understand what is being compared to in situ data." Response: We have updated the description of virtual stations in the introduction ( page 3 line 20-22) to read as follows: "VS are locations where ground tracks of exact repeat altimetry mission orbits cross rivers, enabling the development of a time series of water elevation observations. VS can be thought of much in the same way as an in situ river gaging station, but are entirely derived from remote sensed measurements of river surface elevation." We have placed this text as its own paragraph in the introduction to allow readers to find it more easily.

Specific comments: 1. "Page 1 Line 20: in "either quantitatively for VS where : : :" should it not be "VS levels where : : :.""

Response: We have changed this sentence (Page 1 Line 20) for clarity based on the reviewer's suggestion. It now reads as follows:

"We evaluated every VS, either quantitatively for VS locations where in-situ gages are available, or qualitatively using a grade system."

2. "Page 1 Line 21: As these are multiple VS, should this be VSs? I also wonder about

the grammatical use of VS throughout the paper."

Response: While there is some precedent for using VS for both the singular and plural of virtual station, within the river radar altimeter literature, we agree with the author the using VSs for the plural from increases readability. We have changes 43 instances of VS to VSs throughout the document.

3. "Page 3 Line 27: This is the first instance of VS in the main Text , please spell it out."

Response:

This line of text was corrected to spell out VS and is now included in a separate paragraph ahead of its previous location as a part of our response to specific comment 3.

4. Page 3 Line 28: Same with GRRATS, and with all the several acronyms throughout the paper.

Response:

We changed this use of GRRATS (now Page 3 Line 20) to read as follows:

"Virtual Stations (VSs) are the fundamental organizational element for the Global River Radar Altimeter Time Series (GRRATS), as well as other altimetry datasets for rivers."

Additionally corrections were made for the following acronyms.

Page 3 Line 24 changed to "Database for Hydrological Time Series over Inland Waters (DAHITI)"

Page 3 Line 25 changed to "River&Lake Near Real Time (NRT)"

Page 4 Line 30 changed to "Global River Widths from Landsat (GRWL)"

Page 5 Line 7 changed to "Earth Gravitational Model 2008 (EGM08)"

Page 5 Line 9 changed to "Digital Elevation Model (DEM)"

Page 5 line 11-12 changed to "Shuttle Radar Topography Mission (SRTM), Global Multi-Resolution Terrain Elevation Data (GMTED), and Advanced Spaceborne Thermal Emission and Reflection Radiometer (ASTER),"

5. Page 4 Line 29: Its unclear where this polygon came from. Would calling it the 'mask' be more informative, as it was earlier? How was this polygon defined? This is not described in the text.

Response:

We have changed polygon to mask, and included a bit more description of mask construction. It now reads as follows:

"We extracted altimeter observations at the VS from the GDRs; this consisted of three steps. First we spatially joined Landsat imagery (selected from times of mean river discharge) compiled for the Global River Widths from Landsat (GRWL) river centerlines dataset (Allen & Pavelsky, 2015; Allen & Pavelsky, 2018) with satellite ground tracks to define the width extent of the mask used for the extraction of water elevations. Each mask was constructed using the width extent and upstream and downstream limits that were 2km perpendicular to the crossing location."

6. "Page 6 Line 20: Maybe rename this section 'Data Description'"

Response:

We prefer to retain the name 'Data Evaluation". This sections describes how we went about evaluating each VS to create the evaluation data included with the final product.

7. "Page 6 Line 21: is this a typo: '50.M'?"

Response:

Corrected the above to read 50M.

8. "Page 9 Line 16: The authors note there are two streams illustrated in Figure 4, but

there are three."

Response:

We have edited this section to read as follows:

"Therefore, we compared three VS locations that are in each of the four datasets discussed (one on the Amazon, Congo, and Brahmaputra). Figure 4 (a-c) show time series anomaly at each VS and the closest gage. Note that time series lengths are limited to the shortest time series in the comparison and do not match the coverage of any particular mission, and that River&LakeNRT data was unavailable for the VS location on the Brahmaputra."

9. "Figure 1: Is the black background really the best choice?"

Response:

We tried quite a few options in preparing this figure, and have consistently found that black does the best job providing a contrasting color for the two different color schemes presented in this figure.

10: Figure 4: "A legend would be very helpful. Same in Figure 5."

Response:

We have added a legend to Figure 4 and updated the caption with the following:

"Note that the legend in panel (b) apples to all of figure 4."

We have also added a legend to each panel of figure 5.

Please also note the supplement to this comment:
https://www.earth-syst-sci-data-discuss.net/essd-2019-84/essd-2019-84-AC1-supplement.pdf

2019.

---

## Referee Comment (RC2) · Anonymous Referee #2 · 5 Oct 2019

The authors have done tremendous work in compiling a data set of global river water levels based on radar satellite altimetry from Jason-2 and Envisat. This data set also includes several arctic rivers, which are absent in other altimetry services. The paper is generally well written and easy to read. However, the sections regarding the evaluation of the data sets need some clarifications. I hope that the authors find the comments to be constructive.

General comments Section 3.3-3.4 describing the evaluation of the data set needs to be presented more clearly. These sections contain a lot of numbers in the text, which makes it difficult to read. I recommend putting the summary statistics in tables instead.

[Figure]

I find figure 3 confusing. An improvement could be to have a column of plots for each summary statistics

Since this paper describes a data product I would expect to have a description of a data file/product. The description could just be put in an appendix. Please also expand section 4. Hence, add some general information about the product, and the map webpage.

On the interactive map, the individual time series are shown with error bars. It is however not clear what these error bars represent. This is important for the user when applying the data. Please make this clear in the paper. Please also add error bars to the time series in figure 2.

Specific comments:

p2,l9-10: What is meant by this sentence " Newer radar altimeter missions like Sentinel-3 are improving the contemporary record with features like automated processing. "

p2,l28: "be accurately be measured" -> "be accurately measured"

p5, l27-28: Maybe the authors could add approximately dates for the winter period

p6, l7-8: "Analyses showed that VS-stream gage distance was often not an accurate predictor of height anomaly differences". please, clarify/comment on this statement. Why should the satellite data have a better fit for a station farther away? Why not look at the resemblance among the gauges along the river, this will give you an idea.

p6,l25-28: The description of your qualitative evaluation is a bit vague. And in the result section, you do not present any summary measure. Hence, what are the limits for getting grades A, B, C or D? An alternative measure could be to evaluate the along-track height variation. Please expand your description.

p9, l25: Please, clarify what is meant by this: "We suggest that individual VS data point error be estimated as the STDE of the time series they are a component of."

Figure 1: The red color in the two evaluations is a bit similar. Maybe make the colors more different or make two figures.

Figure 2: Please add error bars and describe what they represent.

Figure 3: Reorganize for clarity. Make the font of the axis and labels larger.
* * *

---

## Author Comment (AC3) · 10 Oct 2019

We would like to thank the reviewer for the time and effort they have put into reviewing our manuscript, as well as for the helpful comments they have provided. Reviewer comments are shown below in italics, and our response in plain type. We have numbered and labeled the reviewer's comments, "General Comment 1", etc. The authors have done tremendous work in compiling a data set of global river water levels based on radar satellite altimetry from Jason-2 and Envisat. This data set also includes several arctic rivers, which are absent in other altimetry services. The paper is generally well written and easy to read. However, the sections regarding the evaluation of the

data sets need some clarifications. I hope that the authors find the comments to be constructive. General comment 1 Section 3.3-3.4 describing the evaluation of the data set needs to be presented more clearly. These sections contain a lot of numbers in the text, which makes it difficult to read. I recommend putting the summary statistics in tables instead. Response: We agree that a table will be useful for aiding readers in understanding our validation process and results. We have added table 1 and referenced it in the beginning of section 3.3. We carefully reviewed the text and found that all of the sentences are important to the paper, so have kept this section as is. We think having the Table to refer to will aid in reading these sections as well. General comment 2 I find figure 3 confusing. An improvement could be to have a column of plots for each summary statistics Response: Based on the reviewer's recommendation, we have updated figure 3 so that each fit statistic is in its own column. In order to make the spacing of the figure less awkward, we removed NSE plots that included values <0. We have updated the text in in section 3.3 to reflect the new figure labels. We have also updated the figure caption to describe the amount of data missing in figures 3a-c. It reads as follows: Figure 3 Virtual Station fit statistics computed with all available evaluation gages located in the same river and closest comparison. Please note that NSE values are plotted here only when greater than 0 to enable readers to more easily see the majority of the data. 12.2%, 28.8% and 17.2% of the total data are not shown in panel a, b, and c respectively. Panel (a) histogram of the max NSE >0 at each VS in the dataset, Panel (b) histogram of the median NSE > 0 at each VS in the dataset, Panel (c) histogram of closest NSE>0 , Panel (d) A histogram of the minimum STDE in the dataset, Panel (e) A histogram of the median STDE all the VSs in the dataset, Panel (f) histogram of closest STDE, Panel (g) histogram of the max R at each VS in the dataset, Panel (h) histogram of the median R at each VS in the dataset, Panel (i) histogram of closest R. The new figure is attached to this comment.

General comment 3 Since this paper describes a data product I would expect to have a description of a data file/product. The description could just be put in an appendix. Please also expand section 4. Hence, add
some general information about the product, and the map webpage. Response: We added an appendix with a file description and updated section 4 to read as follows: GRRATS (DOI 10.5067/PSGRA-SA2V1) is available at https://podaac.jpl.nasa.gov/dataset/PRESWOT_HYDRO_GRRATS_L2_VIRTUAL_STATION_HEIGHTS_V1for non-commercial use only (Durand et al., 2016). Data are provided in NETCDF format. For a file content description please see Appendix A. An interactive map of the data is located at http://research.bpcrc.osu.edu/grrats/. This tool is intended for exploration only, and may not reflect the most up-to-date version of the data. As with figure 2, error bars represent the range of the values that were averaged to generate each data point (does not include filtered data points).

General comment 4 On the interactive map, the individual time series are shown with error bars. It is however not clear what these error bars represent. This is important for the user when applying the data. Please make this clear in the paper. Please also add error bars to the time series in figure 2. Response: We have added error bars to figure 2 and described them in the figure 2 caption, the text of section 3.2 and the text of section 4 with the following description: Error bars represent the range of the values that were averaged to generate each data point (does not include filtered data points). Data necessary to compute error bars are a part of the data product. We updated the figure 4 caption with the following to explain the lack of error bars: GRRATS error bars are not shown to improve readability.

Specific comment 1 p2,l9-10: What is meant by this sentence " Newer radar altimeter missions like Sentinel- 3 are improving the contemporary record with features like automated processing. " Response: Updated to read as follows: "Newer radar altimeter missions like Sentinel-3 are improving the contemporary record with features like automated processing, alleviating the need for retracking and other post processing to generate useful measurements." Specific comment 2 p2,l28: "be accurately be measured" -> "be accurately measured" Response: Changed as requested. Specific comment 3 p5, l27-28: Maybe the authors could add approximately dates for the

winter period Response: Updated to read as follows: "If ice breakup data were not available, we applied broad date limits regionally, using observations from the Pavelsky and Smith (2004) study of Arctic river ice breakup timing. Breakup dates range from late September to early June." Specific comment 4 p6, l7-8: "Analyses showed that VS-stream gage distance was often not an accurate predictor of height anomaly differences". please, clarify/comment on this statement. Why should the satellite data have a better fit for a station farther away? Why not look at the resemblance among the gauges along the river, this will give you an idea.

Response: We updated this section to read as follows:

"Analyses showed that VS-stream gage distance was often not an accurate predictor of height anomaly differences. This is likely due to the hydraulics (width, nearby dams, confluences) of a more distant gage being more similar to the location of the VS than the most proximal gage."

We believe that a detailed study of river-by-river comparison of in situ gage observations is out-of-scope for this manuscript.

Specific comment 5 p6,l25-28: The description of your qualitative evaluation is a bit vague. And in the result section, you do not present any summary measure. Hence, what are the limits for getting grades A, B, C or D? An alternative measure could be to evaluate the alongtrack height variation. Please expand your description. Response: We added the following to this section: "Letter grades are take in to consideration all of these criteria, but in general, VSs with an A rating would have 1 or fewer obvious outliers per year, no more than 2 cycles filtered out per year, and will fit nicely above VS downriver and below VS upriver. A D rating might be applied to a VS with 3 or more outliers per year, 5 or more cycles missing per year, and might fall below VS downriver from it, and above VS upriver from it." Additionally, we added a table with the letter grade results and referenced it in the results section (table2). P7, l6-7 "We gave the remaining 27 rivers qualitative letter grades based on number of missing

data points, obvious outliers, and agreement with nearby stations. These grades are included with the data for end users (Table 2). The majority of rivers evaluated this way fall into the B or C category (∼61%), with only ∼15% getting an A rating" Specific comment 6 p9, l25: Please, clarify what is meant by this: "We suggest that individual VS data point error be estimated as the STDE of the time series they are a component of." Response: Updated to the following for clarity: "While we do not provide error estimates at the individual data point level, we suggest that individual VS data point error be treated as the STDE of the time series they are a component of." Specific comment 7 Figure 1: The red color in the two evaluations is a bit similar. Maybe make the colors more different or make two figures. Response: This is a good point: the colors are similar. However, note that each river is evaluated entirely qualitatively or quantitatively. Also note that the qualitative (cool) and quantitative (warm) color maps are easily distinguished. Thus, the similarity of the colors does not affect readers' ability to interpret the figure. To prevent confusion, we have added the following to the figure caption: "Each river is evaluated using only one of these methods." Specific comment 8 Figure 2: Please add error bars and describe what they represent. Response: Error bars are added and described in the figure caption, as well as the main text. Specific comment 9 Figure 3: Reorganize for clarity. Make the font of the axis and labels larger. Response: Figure reorganized based on the reviewer's recommendation. Font size increased.

Please also note the supplement to this comment:
https://www.earth-syst-sci-data-discuss.net/essd-2019-84/essd-2019-84-AC3-supplement.pdf

[Figure]

**(a)**
count
Max NSE>0

**(d)**
count
Min STDE (m)

**(g)**
count
Max R

**(b)**
count
Median NSE>0

**(e)**
count
Median STDE (m)

**(h)**
count
Medain R

**(c)**
count
Closest NSE >0

**(f)**
count
Closest STDE (m)

**(i)**
count
Closest R

**Fig. 1.**

---

## Author Response (AR2)

We would like to thank the editor for the time and effort they have put into reviewing our manuscript, as well as for the helpful comment they have provided. The editor's comment is shown below in italics, and our response in plain type.

Comment:

*I only have one suggestion. I thank the authors for adding detail on what a virtual station is. The location in which this has been inserted feels like it has just been plopped into the text. Could I suggest that content be moved so that the paragraph page 4 near line 10 reads: ".... opportunistic exploitation of virtual stations (VSs) on the world's largest rivers specifically suited for the needs of global hydrological applications. GRRATS uses the VS as its fundamental organizational element. VSs are locations where ground tracks of exact repeat altimetry mission orbits cross rivers, enabling the development of a time series of water elevation observations. VSs can be thought of much in the same way as an in situ river gaging station, but are entirely derived from remote sensed measurements of river surface elevation. GRRATS is an "Earth.....".*

*The paper is good technically, and I was able to access the data.*

Response:

We agree with the editor's assessment and have made the suggested change.

Marked up manuscript:

[revised manuscript text omitted]